# Molecular Epidemiology of Rotavirus A in Calves: Evolutionary Analysis of a Bovine G8P[11] Strain and Spatio-Temporal Dynamics of G6 Lineages in the Americas

**DOI:** 10.3390/v15102115

**Published:** 2023-10-19

**Authors:** Enrique L. Louge Uriarte, Alejandra Badaracco, Maximiliano J. Spetter, Samuel Miño, Joaquín I. Armendano, Mark Zeller, Elisabeth Heylen, Ernesto Späth, María Rosa Leunda, Ana Rita Moreira, Jelle Matthijnssens, Viviana Parreño, Anselmo C. Odeón

**Affiliations:** 1Instituto Nacional de Tecnología Agropecuaria, Instituto de Innovación para la Producción Agropecuaria y el Desarrollo Sostenible, Ruta 226, km 73.5, Balcarce B7620, Buenos Aires, Argentina; leunda.maria@inta.gob.ar (M.R.L.); anarita54@yahoo.com.ar (A.R.M.); 2Instituto Nacional de Tecnología Agropecuaria, EEA Montecarlo, Av. El Libertador Nº 2472, Montecarlo CP3384, Misiones, Argentina; badaracco.alejandra@inta.gob.ar; 3Facultad de Ciencias Veterinarias, Departamento de Fisiopatología, Centro de Investigación Veterinaria de Tandil (CIVETAN), Universidad Nacional del Centro de la Provincia de Buenos Aires, Paraje Arroyo Seco s/n, Tandil CP7000, Buenos Aires, Argentina; mspetter@vet.unicen.edu.ar (M.J.S.); armendano.joaquin@gmail.com (J.I.A.); 4Instituto Nacional de Tecnología Agropecuaria, EEA Cerro Azul, Ruta 14, km 836, Cerro Azul CP3313, Misiones, Argentina; mino.samuel@inta.gob.ar; 5Laboratory of Viral Metagenomics, Department of Microbiology, Immunology and Transplantation, Rega Institute, University of Leuven, Herestraat 49, 3000 Leuven, Belgium; mark.zeller@uz.kuleuven.ac.be (M.Z.); elisabeth.heylen@rega.kuleuven.be (E.H.); 6Facultad de Ciencias Agrarias, Universidad Nacional de Mar del Plata, Ruta 226, km 73.5, Balcarce B7620, Buenos Aires, Argentina; ejaspath@gmail.com (E.S.); aodeon@mdp.edu.ar (A.C.O.); 7Instituto Nacional de Tecnología Agropecuaria, Instituto de Virología e Innovaciones Tecnológicas, Nicolas Repetto y de los Reseros s/n, Hurlingham CP1686, Buenos Aires, Argentina

**Keywords:** calves, diarrhea, rotavirus, genotypes, G8-lineages, phylodynamics, phylogeography

## Abstract

Rotavirus A (RVA) causes diarrhea in calves and frequently possesses the G6 and P[5]/P[11] genotypes, whereas G8 is less common. We aimed to compare RVA infections and G/P genotypes in beef and dairy calves from major livestock regions of Argentina, elucidate the evolutionary origin of a G8 strain and analyze the G8 lineages, infer the phylogenetic relationship of RVA field strains, and investigate the evolution and spatio-temporal dynamics of the main G6 lineages in American countries. Fecal samples (*n* = 422) from diarrheic (beef, 104; dairy, 137) and non-diarrheic (beef, 78; dairy, 103) calves were analyzed by ELISA and semi-nested multiplex RT-PCR. Sequencing, phylogenetic, phylodynamic, and phylogeographic analyses were performed. RVA infections were more frequent in beef (22.0%) than in dairy (14.2%) calves. Prevalent genotypes and G6 lineages were G6(IV)P[5] in beef (90.9%) and G6(III)P[11] (41.2%) or mixed genotypes (23.5%) in dairy calves. The only G8 strain was phylogenetically related to bovine and artiodactyl bovine-like strains. Re-analyses inside the G8 genotype identified G8(I) to G8(VIII) lineages. Of all G6 strains characterized, the G6(IV)P[5](I) strains from “Cuenca del Salado” (Argentina) and Uruguay clustered together. According to farm location, a clustering pattern for G6(IV)P[5] strains of beef farms was observed. Both G6 lineage strains together revealed an evolutionary rate of 1.24 × 10^−3^ substitutions/site/year, and the time to the most recent common ancestor was dated in 1853. The most probable ancestral locations were Argentina in 1981 for G6(III) strains and the USA in 1940 for G6(IV) strains. The highest migration rates for both G6 lineages together were from Argentina to Brazil and Uruguay. Altogether, the epidemiology, genetic diversity, and phylogeny of RVA in calves can differ according to the production system and farm location. We provide novel knowledge about the evolutionary origin of a bovine G8P[11] strain. Finally, bovine G6 strains from American countries would have originated in the USA nearly a century before its first description.

## 1. Introduction

Neonatal calf diarrhea (NCD) is a leading cause of disease and death in dairy and beef calves [1]. It generates important economic losses to the livestock industry due to poor growth, death, and expenses in treatments [1,2]. Also, in cow-calf beef farms, treating diarrheic calves is labor-intensive and physically risky [1]. The etiology of NCD is complex and involves enteric pathogens (virus, bacteria, and protozoa) along with predisposing factors such as the environment, immunity, and nutrition [2,3,4].

Rotaviruses (RVs) belong to the family *Sedoreoviridae* and genus *Rotavirus*. Currently, the International Committee on Taxonomy of Viruses (ICTV) recognizes nine species (groups) that are designated RVA to RVD and RVF to RVJ [5]. The serological cross-reactivity and amino acid (aa) sequence identity of VP6 as well as the host ranges allow for us to distinguish these species [6,7]. RVA is not only the main pathogen associated with NCD [2] but also the most clinically relevant species infecting other ruminants [3]. Previous studies in different countries report a variable frequency of RVA infections in diarrheic calves, ranging from 27% to 79.9% [8,9,10,11,12,13,14]. Aside from other factors, virus detection methods and management practices adopted in cattle farms may explain this variability. Although subclinical infections contribute to environmental contamination [3], few studies have compared RVA infections in diarrheic and non-diarrheic (subclinical) calves from beef and dairy farms [8,13,14].

The RVAs genome consists of 11 double-stranded RNA (dsRNA) segments surrounded by a non-enveloped, triple-layered, protein capsid [15]. These gene segments range in size from 667 to 3302 nucleotides (nt) and carry open reading frame (ORF) sequences coding six structural viral proteins (VP1-4, VP6, VP7) and five to six non-structural proteins (NSP1-NSP6) [3,7]. RVA strains are classified into G (glycoprotein) and P (protease-sensitive protein) genotypes and serotypes based on the genetic and serologic specificities of the outer capsid proteins, VP7 and VP4, respectively [15]. Thus far, RVA strains carrying 14 G genotypes (G1–6, G8–10, G12, G15, G18, G21, and G24) and 12 P genotypes (P[1], P[5]–[8], P[11], P[17], P[21] P[23], P[29], P[33], P[35], and P[38]) have been detected in cattle [3]. Of all these genotypes, G6P[5], G6P[11], and G10P[11] are the most prevalent G/P combinations [16]. In contrast, the G8 genotype, regardless of the combined P genotype, is by far less frequent [11,17,18,19], or not detected at all in many countries [20,21,22,23,24,25]. Both VP7 and VP4 proteins elicit neutralizing antibody responses [15], and vaccines given to pregnant dams before calving significantly reduce the incidence of diarrhea and mortality in neonatal calves through passive immunity [26]. The continuous surveillance of G/P types among circulating strains gives key information to develop vaccines [23]. Genotyping tools are therefore useful for this purpose and track the epidemiological pathways of RVs transmission among host species [7].

The whole genome classification system of RVA strains is based on nt percentage identity cutoff values, which allow for us to define the genotypes for all the 11 gene segments. Thus, the schematic nomenclature Gx-P[x]-Ix-Rx-Cx-Mx-Ax-Nx-Tx-Ex-Hx (x = Arabic numbers) denotes the genotypes for the VP7-VP4-VP6-VP1-VP2-VP3-NSP1-NSP2-NSP3-NSP4-NSP5/6 coding genes, respectively [27]. This classification enables us to elucidate the most likely origin and trace the evolutionary patterns of animal and human strains [28]. Also, this approach is very important because interspecies transmission and reassortment events are major ways to increase the genome diversity of RVA strains [3,29].

G8 RVA strains are particularly interesting for many reasons: they infect diverse mammalian species from the order Artiodactyla [29,30]; show zoonotic potential for direct transmission [31,32,33]; and can reassort single or multiple gene segments with human strains to generate partial bovine-like (or other artiodactyl) genotype constellations [34,35,36]. Although the first G8 RVA strain (NCDV-Cody) infecting calves was reported 54 years ago in the USA [37], as of today only a handful of complete genomes of bovine G8 strains have been genetically characterized in South Africa [29], Nigeria [33], Thailand, Japan [38], and Brazil [39]. The Brazilian bovine G8P[11] strain (Y136) was phylogenetically analyzed only for VP7. Exhaustive phylogenetic analyses are necessary to determine the most likely origin and genetic relatedness to other G8 strains [31,32]. Moreover, the diversifications of VP7 sequences among G8 strains is increasing, and six lineages have been proposed for this genotype [36,38,40].

The phylogenetic relationship of bovine RVA field strains has been investigated using the VP7 and VP4 (VP8* subunit) coding genes [20,21,22,23,38,41]. However, the phylogenetic linkages of circulating strains according to the production system and farm location has not been evaluated. Regarding the molecular epidemiology of bovine RVA, some studies have assessed the G6 lineages in field strains. Among them, G6 lineage III [G6(III)], G6 lineage IV [G6(IV)], and G6 lineage V [G6(V)] were the most prevalent [13,21,23,25,41,42]. In American countries, the evolution and origin of G6(III) (i.e., Hun 4-like) [43] and G6(IV) (i.e., NCDV-like) [44] lineage strains are unknown. The first bovine RVA strain, designated NCDV-Lincoln (G6P[1]), was reported in the USA in 1969 [45] and belonged to the G6(IV) lineage [44].

Continuous knowledge of the epidemiology and genetic diversity of RVA is essential to implement preventive measures [16]. The aims of this study were as follows: (a) to compare the frequency of RVA infections and the distribution of G/P genotypes in diarrheic and non-diarrheic calves from beef and dairy farms in major livestock regions of Argentina; (b) to elucidate the evolutionary origin of an unusual G8 strain and analyze the G8 lineages; (c) to infer the phylogenetic relationship of RVA study strains and assess the influence of the production system and the location of farms on their relationship; (d) investigate the evolution and spatio-temporal distribution of the major G6 lineages in American countries. The resulting data of this study show differential trends when comparing the infection frequency, G/P genotypes, genetic diversity, and phylogeny of RVA strains between calves from beef and dairy farms. Our research provides novel knowledge about the evolutionary origin of bovine G8 strains. Finally, in American countries, bovine G6 strains would have originated in the USA nearly a century before its first description in diarrheic calves from this country.

## 2. Materials and Methods

### 2.1. Farms, Calves, and Sample Collection

From September 2007 to December 2010, a total of 32 farms were selected by convenience (non-probability method) [46], because veterinary practitioners requested to detect enteric pathogens during NCD outbreaks. The farms were located in Buenos Aires province, where the largest cattle population of Argentina is reared (14,883,528 out of 40,023,083; ~37% of all cattle) [47]. Beef farms (*n* = 14) were located in eight districts inside (7/8) or just outside (1/8) the beef production region of “Cuenca del Salado and Depresion de Laprida” (Appendix A). This region accounts for 48% of the cattle population in Buenos Aires province and is the most important production region in Argentina [48]. Geographically, this region can be split into “Cuenca del Salado” and “Depresion de Laprida” [49]. Dairy farms (*n* = 18) were located in 10 districts and three dairy regions, namely “Cuenca Mar y Sierras” (*n* = 14), “Cuenca Oeste” (*n* = 3), and “Cuenca Abasto Norte” (*n* = 1) [50]. Three districts included both beef and dairy farms (Appendix A).

In beef farms, calves were nursed by cows (cow-calf pairs) in seeded and native pastures until weaning at 5–8 months of age. In dairy farms, calves were reared using individual stakes (83.3%, 15/18 farms), hutches (5.6%, 1/18), or both (11.1%, 2/18) until weaning at 2 months of age. On average, sampled calves were 14 days old (range, 2 to 69 days). The feed supply included milk (55.6%, 10/18), milk replacers (22.2%, 4/18), milk plus milk replacers (16.7%, 3/18), whey plus milk (5.5%, 1/18), and energy-protein concentrates (83.3%, 15/18). The vaccination of the dams to prevent NCD was conducted in 86% (12/14) of beef and 55.5% (10/18) of dairy farms (Appendix A).

Individual fecal samples (*n* = 422) were collected from diarrheic (beef, *n* = 104; dairy, *n* = 137) and non-diarrheic (beef, *n* = 78; dairy, *n* = 103) calves during NCD outbreaks (Appendix A). The fecal consistency was scored at the moment of sample collection as follows: 0 (normal), 1 (pasty), 2 (loose feces), and 3 (watery feces); fecal scores of 2 and 3 were classified as diarrheic [51]. Details on fecal scores among beef and dairy calves are shown in Figure 1. Fecal samples collected per farm (farm sample) averaged 13 either in beef (range, 3–28 calves) and dairy (range, 3–22 calves) farms (Appendix A). Overall, the proportions of diarrheic (beef, 57.1%; dairy, 57.1%) and non-diarrheic (beef, 42.9%; dairy, 42.9%) calves were identical in both production systems; in each farm sample at least 20% of calves were diarrheic. Feces were transported in ice-cooled containers and stored at −20 °C until analysis [52].

One dairy farm, designated DR, was visited in 2009, 2010, and 2011 due to severe NCD outbreaks. On this farm, the fecal samples collected in 2009 were included for statistical, genotype, and phylogenetic analyses, whereas the samples collected in 2010 and 2011 were included for genotype comparisons and phylogenetic analysis.

### 2.2. RVA Reference Strains and Antigen Detection

The reference strains Indiana (G6(IV)P[5]), B223 (G10P[11]), and NCDV-Cody I801 (G8P[1]), as well as a bovine field strain (G6(III)P[11]) from Argentina, whose VP7 and VP8* genotypes were previously confirmed by nucleotide sequencing [41], were included as positive controls for antigen-capture, enzyme-linked, immunosorbent assay (Ag-ELISA) and genotyping assays. The tissue culture-adapted reference strains were kindly provided by Dr. Linda J. Saif (FAHRP, The Ohio State University, USA).

Fecal suspensions were prepared as 10% (*w*/*v*) dilutions in PBS-Tween_20_ 0.05% (PBS 1× pH: 7.4), centrifuged (5000× *g* for 5 min), and the supernatant was used to detect RVA by Ag-ELISA as previously described [11]. Fecal samples with insufficient material were only analyzed by Ag-ELISA.

### 2.3. RNA Extraction and G/P Genotyping

To conduct the multiple based genotyping of RVA field strains in Ag-ELISA positive samples, dsRNA was extracted from fecal suspensions with TRIzol (Invitrogen, Carlsbad, CA, USA) following the manufacturer’s instructions. The dsRNA pellet was air dried and resuspended in 50 µL of nuclease-free water (Biodynamics, Buenos Aires, Argentina).

Two semi-nested multiplex (SnM) RT-PCR assays were conducted to identify the G genotypes and G6 linages [G6(III), G6(IV), G8, and G10], or the P genotypes (P[1], P[5] and P[11]) of VP7 and VP8*, respectively [11]. The G-genotyping assay (2nd round of PCR) also avoids the mistyping of G6(III) lineage strains as G8 or G10 [11]. The P-genotyping assay (2nd round of PCR) enables us to identify the most prevalent P genotypes in cattle [16] (Appendix A). The PCR products were resolved in 1.8% agarose gels, stained with ethidium bromide (0.5 μg/mL), and visualized under UV light. Fecal samples displaying different G and/or P genotypes (multiple bands in the SnM RT-PCR) were designated mixed genotypes, whereas farm samples in which ≥1 calf carried mixed genotypes were designated mixed RVA outbreaks.

### 2.4. Sequencing of VP7 and VP8*

Ag-ELISA positive samples from diarrheic calves were selected according to the following criteria: single G and/or P genotype in the sample, production system (beef or dairy), and farm location in production regions. The RT-PCR products of VP7 (1062 bp) and VP4 (VP8* subunit, 877 bp) genes were purified using illustra ExoProStar (GE Healthcare, Little Chalfont, UK). Then, sequencing by the dideoxy chain termination method was conducted bi-directionally with generic primers [11]. This analysis was performed on an automated DNA sequencer ABI 3500 (Applied Biosystems, Foster city, CA, USA) at the Genomic Unit of the Biotechnology Institute (CICVyA, INTA, Hurlingham, Argentina).

### 2.5. Sequencing of RVA Gene Segments

For the full genome sequencing of a selected RVA strain, dsRNA was extracted from the fecal suspensions of an Ag-ELISA positive sample (4385) using the QIAamp Viral RNA Mini Kit (Qiagen/Westburg, Leusden, The Netherlands). The dsRNA gene segments were reverse transcribed and amplified using the Qiagen OneStep RT-PCR kit (Qiagen/Westburg, Leusden, The Netherlands), with primers and cycling conditions described previously [53,54]. Briefly, the stages of RT (50 °C/30 min) and PCR activation (95 °C/15 min) were followed by 35 cycles of amplification: (i) 94 °C/30 s, 50 °C/30 s, and 72 °C/4 min for VP1-VP4 genes; (ii) 94 °C/30 s, 45 °C/ 30 s, and 72 °C/3 min for VP7, VP6, and NSP1-NPS5 genes. Both cycle conditions were followed by a final extension stage (72 °C/10 min). The RT-PCR products were purified with the MSB Spin PCRapace kit (Invitek, Berlin, Germany) and sequenced by using the BigDye Terminator Cycle Sequencing Reaction kit (Applied Biosystems, Foster city, CA, USA) on an automated sequencer ABI PRISM 3130 (Applied Biosystems). Sequencing reactions were conducted bi-directionally with the primers used for each RT-PCR (primers available upon request). Primer walking sequencing with internal primers was performed to cover the ORF sequences of the longest genes (VP1–VP4). This study was conducted at Dr. Jelle Matthijnssens laboratory at the Rega Institute for Medical Research, University of Leuven, Belgium.

### 2.6. Sequence Analysis, Genotype Assignment, and Percentage Identity

Sequencing electropherograms were inspected and corrected with the ChromasPro V1.5 software (Technelysium Pvt. Ltd., 2009). To obtain the ORF sequences of all the 11 gene segments, the nt sequences generated with forward, reverse, and internal primers were aligned and assembled using BioEdit v7.0.2.5 [55]. These sequences were evaluated to accomplish the guidelines proposed by the Rotavirus Classification Working Group (RCWG) [56]. The genotype for each gene segment was assigned using the Rotavirus A Genotype Determination Tool, which was available in the Virus Pathogen Database and Analysis Resource (ViPR) [57]. Currently, this tool has migrated to the Bacterial and Viral Bioinformatics Resource Center.

The Nucleotide BLAST tool (https://blast.ncbi.nlm.nih.gov/Blast.cgi; accessed on 20 March 2023) was run to retrieve similar sequences deposited in the GenBank database. Particularly, the nt sequences of G8 and/or P[11] strains were collected if their backbone genotype constellations were reminiscent to that of the study strain. Multiple sequence alignment files were prepared using ClustalW with default settings, as integrated in the MEGA 7.0 software [58]. These sequence datasets comprised complete and partial ORF sequences of each gene segment. Identity matrices, prepared with the aligned nt and deduced aa sequences, were calculated using the p-distance algorithm [30,59] and the pairwise distance analysis [28] implemented in MEGA 7.0.

### 2.7. Substitution Models and Phylogenetic Analyses

The best-fitting model of nt substitution was evaluated for each sequence dataset using the ModelFinder module [60] available in IQ-TREE v1.6.12 [61]. The substitution models with the lowest Bayesian Information Criterion (BIC) values were chosen, as follows: GTR + F + G4 (VP1, VP2, VP7-G8, NSP2, NSP3), GTR + F + I + G4 (VP3, VP4, VP6, NSP1), TPM2 + F + G4 (NSP4, VP8*-P[11]), TIM2 + F + I + G4 (NSP5), HKY + F + I + G4 (VP7-G6), TPM3 + F + G4 (VP7-G10), and TN + F + G4 (VP8*-P[5]). Phylogenetic trees were constructed using the maximum likelihood (ML) method implemented in IQ-TREE and were visualized with MEGA 7.0. The reliability of the branches was statistically estimated using the ultrafast bootstrap (UFBoot) method [62] with 10,000 replicates. The internal branch nodes with UFBoot values ≥ 90% are highlighted in the phylograms.

To infer the phylogeny of G6, P[5], and P[11] RVA strains according to the production system and the location of farms in Buenos Aires province, additional ORF sequences corresponding to VP7/VP8* (nine strains), VP7 (11 strains), and VP8* (12 strains) were included from previous studies in Argentina [41,44]. Subsequently, phylogenetic reconstruction was carried out using the Kimura 2-parameter substitution model and the Neighbor-Joining (NJ) method as implemented in MEGA 7.0 [28,41].

### 2.8. Determination of G8 Lineages

The VP7-G8 genotype dataset included 183 complete and partial (≥73% completeness) ORF sequences. This dataset, which was designated RVA-G8(lineages) (see Appendix A), did not include sequences of RVA strains with 100% identity and obtained from the same host species and/or country. The phylogenetic tree was inferred using the Kimura 2-parameter substitution model and the NJ method [28,41]. Thereafter, a cutoff value to best discriminate G8 lineages was estimated. Briefly, a matrix of genetic distance values was prepared using the pairwise distance algorithm and the p-distance model available in MEGA 7.0 [58]. The pairwise distance frequency graph was created by plotting the pairwise distance values, expressed as percentages, in the abscissa (x-axis) and their frequencies in the ordinate (y-axis). The genetic distance values between strains of the same lineage were designated “intra-lineage distances”, whereas the genetic distance values between strains of different lineages were designated “inter-lineage distances”. The cutoff value enables to differentiate between the intra- and inter-lineage genetic distances [28,41]. The lineage numbers were assigned in accordance with previous papers [38,40].

### 2.9. Phylodynamic and Phylogeographic Analyses of G6 RVA Strains

Three VP7 sequence datasets [RVA-G6, RVA-G6(III), and RVA-G6(IV)] (see Appendix A) were prepared for evolutionary analyses of bovine G6 RVA strains from American countries. All nt sequences were confirmed as non-recombinant using the Recombination Detection Program (RDP) v4.100 [63]. The presence of substitution saturation was not detected by the Xia’s test implemented in the DAMBE software v7.2.102 [64]. The phylogenetic signal was evaluated by the likelihood mapping method available in IQ-TREE v1.6.12. The phylogenetic noise was low to rather low for each sequence dataset (15.9%, 32.2%, and 7.5%, respectively). A positive correlation between the genetic divergence and sampling time was observed using the Root-to-Tip analysis with TempEst v1.5.3 [65] (see Appendix A).

The evolutionary rate (substitutions/site/year, s/s/y), the time to the most recent common ancestor (TMRCA), and the spatial dynamics were co-estimated using the Bayesian Markov Chain Monte Carlo (MCMC) approach in the BEAST software package v1.8.4 [66], which is available in the CIPRES Science Gateway server [67]. The nt substitution models K3P + I + G4 (RVA-G6), HKY + G4 [RVA-G6(III)], and K3P + G4 [RVA-G6(IV)], flexible models, i.e., the uncorrelated lognormal relaxed molecular clock, and a Bayesian Skyline demographic model were used as coalescent parameters.

The spatio-temporal process was evaluated on time-scaled genealogies over discrete sampling locations (countries) using a Bayesian stochastic search variable selection procedure [68] and an asymmetric model. This analysis was performed in the BEAST package v1.8.4. A Bayes Factor (BF) test was used to weigh the significance of the linkages between countries, using the SpreaD3 software v0.9.6 [69]. BF > 3 was considered a well-supported link.

The MCMC runs were carried out long enough to ensure convergence (effective sample size values > 200), which was checked in Tracer v1.7.1 (BEAST package) [70] after 10% burn-in. The uncertainty of parameter estimates was evaluated by the 95% highest posterior density intervals (95% HPD). The maximum clade credibility tree (MCCT) was annotated using TreeAnnotator (BEAST package) and visualized with Figtree v1.4.4 [70].

### 2.10. GenBank Accession Numbers

The nt sequences were deposited in the GenBank database under the accession numbers OR253956-OR253973 (VP7-G6), OR253974-OR253976 (VP7-G10), OR269758-OR269768 (VP8*-P[5]), OR269769-OR269778 (VP8*-P[11]), OR344101 (VP7-G8), OR344102 (VP4-P[11]), OR344103 (VP6), OR344104-OR344106 (VP1-VP3), and OR344107-OR344111 (NSP1-NSP5).

### 2.11. Statistical Analyses

The statistical analysis was conducted in R v4.2.2 (R Core Team, Vienna, Austria). The data were analyzed using binary and multinomial logistic regression models that were fitted with the R packages ‘survey’ v4.1-1 [71] and ‘svyVGAM’ v1.2 [72], respectively. In binary logistic regression models, bias reduction methods were applied (r package ‘brglm2′ v0.9) [73]. In all the analyses, the year of sampling was included as a stratifying factor and the farm was considered as a clustering factor.

The frequency of RVA infections was compared between production systems (beef or dairy) using binary logistic regression. The analysis was adjusted by the clinical condition of the calves (diarrheic or non-diarrheic). Also, the effect measure modification by clinical condition was assessed in the multiplicative scale by including in the model the product term between production system and clinical condition. The frequency of RVA infections was compared between clinical conditions using binary logistic regression and adjusting the analysis by the production system. The effect measure modification by production system was also assessed in the multiplicative scale following the aforementioned methodology. For this analysis, only the 27 farms with RVA-infected calves were included in order to avoid positivity violations [74]. Also, animals infected with bovine coronavirus (BCoV) (three dairy calves from two dairy farms) were excluded from this analysis. The latter was conducted considering that the interaction between RVA and BCoV might increase the risk of diarrhea and that this interaction could not be addressed properly.

The overall distribution of G and P genotypes or G/P combinations was compared between production systems using multinomial logistic regression. Also, separated binary logistic regression models were used to compare the proportion of each genotype between production systems. The False Discovery Rate correction [75] was applied to account for multiplicity.

## 3. Results

### 3.1. Frequency of RVA Infections

At farm level, RVA infections occurred in 86% of beef (12/14) and 83.3% (15/18) of dairy farms. Therefore, at least one calf was shedding RVA in 84.4% (27/32) of farms. Overall, RVA infections were detected in 17.5% (74/422) of calves. According to the production system, the frequency of RVA infections tended to be higher in beef (22.0%, 40/182) than in dairy (14.2%, 34/240) calves (odds ratio (OR) = 1.75, IC 95%, 0.98–3.13, *p* = 0.058). This association was not modified by the clinical condition (diarrheic or non-diarrheic) of the animal (*p* = 0.636), with RVA infections more frequent in beef farms, either in diarrheic [beef = 31% (32/104); dairy = 21.2% (29/137)] or non-diarrheic calves [beef = 10.2% (8/78); dairy = 4.8% (5/103)]. According to the clinical condition, RVA infections were significantly more frequent in diarrheic (25.3%, 61/241) than in non-diarrheic (7.2%, 13/181) calves (OR = 3.74, IC 95%, 2.26–6.21, *p* < 0.001). This association was not modified by the production system (*p* = 0.949), with RVA infections more frequent in diarrheic calves, either in beef [diarrheic = 32.7% (32/98); non-diarrheic = 11.4% (8/70)] or dairy farms [diarrheic = 23.6% (26/110); non-diarrheic (7.1%, 5/70)].

### 3.2. Genotyping of RVA Field Strains

Both SnM RT-PCR assays allowed us to detect the G and P genotypes in most (*n* = 67) RVA positive samples (91% and 100%, respectively). The G untyped (GX) samples had no detectable PCR products after both rounds. Altogether, the most frequent G genotypes and G6 lineages were G6(IV) (50.7%, 34/67) and G6(III) (22.4%, 15/67) (Appendix A), followed by mixed genotypes of G6(IV) + G10 (8.9%, 6/67). Of note, G8 was detected in one fecal sample from a diarrheic dairy calf. Regarding the P genotypes, the most frequently detected were P[5] (58.2%, 39/67) and P[11] (37.3%, 25/67), whereas mixed genotypes of P[5 + 11] were infrequent (4.5%, 3/67) (Appendix A) and P[1] was not detected.

According to the production systems, a different distribution of G and P genotypes or G/P combinations was observed (*p* < 0.001) (Table 1). The G6(IV)P[5] combination was the most frequent in beef calves (Table 1) and farms with RVA positive calves (91.7%, 11/12). In contrast, the G6(III)P[11] combination was the most frequent in dairy calves (Table 1) and farms with RVA positive calves (40%, 6/15). Mixed genotypes were frequent only in dairy calves (Table 1). Thus, mixed RVA outbreaks occurred more frequently in dairy (40%, 6/15) than in beef (8.3%, 1/12) farms. Regarding the clinical condition, G6(IV)P[5] (46.4%, 26/56), G6(III)P[11] (23.2%, 13/56), G(X)P[5] (7.1%, 4/56), G10P[11] (5.4%, 3/56), G6(III)P[5] (1.8%, 1/56), G8P[11] (1.8%, 1/56), and G(X)P[11] (1.8%, 1/56) combinations, or mixed genotypes (12.5%, 7/56), were identified in diarrheic beef and dairy calves. In contrast, G6(IV)P[5] (63.6%, 7/11), G6(III)P[11] (9.1%, 1/11), and GXP[11] (9.1%, 1/11) combinations, or mixed genotypes (18.2%, 2/11), were identified in non-diarrheic beef and dairy calves.

During annual NCD outbreaks in the dairy farm DR, a total of 45.7% (16/35) calves tested positive for RVA in 2009 (36.8%, 7/19), 2010 (58.3%, 7/12), and 2011 (50%, 2/4). Genotyping assays revealed a single genotype combination per year of sampling, as follows: G6(III)P[11] in 2009, G6(IV)P[5] in 2010, and G10P[11] in 2011 (G10P[11]) (Table 1).

### 3.3. Genome Analysis of the G8P[11] RVA Strain

After G/P genotyping, a G8P[11] strain was detected in one diarrheic dairy calf (4385). As the G8 genotype has been detected sporadically or not at all in molecular epidemiology studies of bovine RVA, this finding prompted us to further investigate the genome of this unusual strain, which was designated RVA/Cow-wt/ARG/4385VT_D_BA/2008/G8P[11] (abbreviated names henceforth, i.e., 4385VT_D_BA). This strain exhibited the genotype constellation G8-P[11]-I2-R5-C2-M2-A13-N2-T6-E12-H3. After BLAST searching, the ORF sequences of all the 11 gene segments showed high genetic similarity (91–99%) with the respective sequences of RVA strains detected in calves from Argentina and Uruguay (VP1, VP4, VP6, and NSP1-NSP5) [14,41,76,77], goat kids from Argentina (VP7) [54], and alpacas (VP2) and guanacos (VP3) from Peru and Argentina, respectively [76,78] (Table 2).

A comparative analysis of genotype constellations and sequence identities was performed across the 11 gene segments of strain 4385VT_D_BA and those of other reference G8 and/or P[11] strains distributed worldwide. In this analysis, the nt and deduced aa sequences of each homologous genotype were compared using the ORF sequences (Table 3). Of note, strain 4385VT_D_BA shared 10 genotypes with the bovine strain B383 (G15P[11]) and nine genotypes with the artiodactyl bovine-like strains Rio_Negro (G8P[1]) and Chubut (G8P[14]), both from guanacos [76], and the caprine bovine-like strain 0040 (G8P[1]) [54]. All of them were reported in Argentina. Taken together, the backbone genotype constellations I2-R2/R5-C2-M2-A3/A11/A13-N2-T6-E2/E12-H3 were conserved among G8 strains obtained from various animal species of the order Artiodactyla, as well as humans, believed to be examples of interspecies transmissions. Particularly, the 11 ORF sequences of strain 4385VT_D_BA were found to share high genetic similarity not only with the aforementioned strains but also with two artiodactyl bovine-like G8 strains, namely C75 (G8P[14]), collected from a vicugna in Argentina [53], and 562 (G8P[14]) and 1115 (G8P[1]), both obtained from alpacas in Peru [78] (Table 3). Phylogenetic studies were conducted to assess the most likely evolutionary origin of the G8 strain under study.

### 3.4. Phylogenetic Analysis of the G8P[11] RVA Strain

The ML trees constructed with the 11 ORF sequences of strain 4385VT_D_BA (details in Table 2) are shown in Figure 2a–f. For the VP7 gene, strain 4385VT_D_BA was most closely related to the caprine bovine-like strain 0040, which both segregated in a subcluster that only included Argentinean artiodactyl bovine-like strains (Figure 2a; % identity in Table 3). Regarding the VP4 gene, strain 4385VT_D_BA was most closely related to the Uruguayan bovine G10P[11] strains LVMS1837 and LVMS2625 (99% nt identity) [14], which grouped inside a large cluster of other Argentinean strains from calves [41] (Figure 2a).

For the VP6-I2 genotype, strain 4385VT_D_BA was segregated in a subcluster that contained four Uruguayan bovine P[11] strains (93% nt identity) (Figure 2b). For the VP1-R5 genotype, strain 4385VT_D_BA was most closely related with the caprine bovine-like strain 0040. Both strains also grouped together with the bovine strain B383 and the artiodactyl bovine-like strain Rio_Negro (Figure 2b; % identity in Table 3). All these strains formed a subcluster of Argentinean R5-carrying strains.

Remarkably, in the phylogeny of the VP2-C2 genotype, strain 4385VT_D_BA clustered most closely with the Peruvian alpaca strains 562 and 1115, as well as with the bovine strain B383 (Figure 2c; % identity in Table 3), rather distantly related to other known C2-carrying RVA strains. For the VP3-M2 genotype, strain 4385VT_D_BA clustered between the Argentinean artiodactyl bovine-like stains Chubut and C75, in one branch, and the bovine strains B383 and Y136 and the caprine bovine-like strain 0040, in another branch (Figure 2c; % identity in Table 3). The strain Y136 (G8P[11]) was previously detected in a Brazilian calf [39].

For the NSP1-A13 genotype, strain 4385VT_D_BA was rather closely related to the bovine strain B383 (% identity in Table 3) and three Uruguayan bovine G10 strains (92% nt identity), which formed a unique subcluster (Figure 2d). Regarding the NSP2-N2 and NSP3-T6 genotypes, strain 4385VT_ D_BA grouped most closely with the Uruguayan bovine G10 strains LVMS2625 and LVMS3053 (97% and 98% nt identity) (Figure 2d), and strain LVMS2625 (99% nt identity) (Figure 2e), respectively.

For the NSP4-E12 genotype, the phylogeny showed a very close relationship between strain 4385VT_D_BA and the Argentinean bovine G6P[5] strain B3700 (99% nt identity), in a cluster with multiple other Argentinian bovine RVA strains [77] (Figure 2e). For the NSP5-H3 genotype, strain 4385VT_D_BA segregated in a subcluster mainly composed of South American bovine P[11] strains, namely strain B383 (% identity in Table 3) and the Uruguayan strains LVMS1788, LVMS3206, LVMS3027, and LVMS3031 (98–99% nt identity). Also, the human bovine-like strain 492SR (G8P[1]), which was reported in a child from Paraguay [79], grouped in this subcluster as well (Figure 2f; % identity in Table 3).

### 3.5. G8 Lineages and Genetic Distance Analysis

Several older studies have suggested six lineages [G8(I)-G8(VI)] within the G8 VP7 genotype [36,38,40]. To update these lineages with the most recent data, phylogenetic and genetic distance analyses were conducted as previously described [41] to evaluate further diversification in this genotype. The RVA-G8(lineages) dataset comprised 184 VP7 sequences of RVA strains from alpacas (1.1% of strains, 2/184), camel (0.5%, 1/184), cattle (17.1%, 28/184), deer (0.5%, 1/184), goats (1.9%, 3/184), guanacos (1.1%, 2/184), humans (71.7%, 132/184), pigs (3.3%, 6/184), sheep (0.5%, 1/184), simians (1.1%, 2/184), and vicugna (0.5%, 1/184), or were collected from the environment (2.7%, 5/184). These strains were reported in Africa (33.7%, 62/184), America (13.6%, 25/184), Asia (35.9%, 66/184), Europe (13.6%, 25/184), and Oceania (3.3%, 6/184). The RVA-G8(lineages) dataset with GenBank accession numbers is available in the Appendix A.

Based on the NJ tree topology, eight lineages [G8(I) to G8(VIII)] were recognized (Figure 3A), and six of them were designated according to previous studies [38,40]. Lineage I consisted of only two G8P[1] strains, one obtained from a calf in North America (NCDV-Cody I801) and the other one isolated from oysters on the coasts of Argentina (Crassostea_ gigas_BsAs). Lineage II was the most diverse because it comprised bovine, bovine-like, and artiodactyl bovine-like strains from many countries and host species (i.e., bovine, deer, human, pig, rhesus macaque, and sheep). These strains also exhibited various P genotypes such as P[1], P[4], P[5], P[7], P[10]-P[12], and P[14]. This lineage included our bovine strain under study (4385VT_D_BA), which was very closely related to Argentinean strains from goats and South American camelids (0040, C75, Rio_Negro, and Chubut) [53,54,76]. Lineage III consisted of human strains from Finland (HAL 1166; G8P[12]), Australia (DG8, BG8.01, and MG8.01; G8P[14]), and India (69M; G8P[10]). Only two bovine strains from South Africa (1604, G8P[1]) and Scotland (678, G8P[5]) grouped in this lineage as well. Lineage IV mainly included human P[4] and P[8] strains from Europe, Africa, and America. Only one bovine strain from Turkey (Amasya-1; G8P[5]) and one camel strain from Sudan (MRC-DPRU447; G8P[11]) clustered in this lineage as well. Lineage V grouped human and bovine G8P[14] strains from Egypt (EGY1850 and EGY229), Hungary (BP1062), India (MP409, 68, 86, and BE4), Taiwan (04-97s379), and Thailand (A5-10). Also, a caprine G8P[1] strain from India (K-98) and human G8P[8] strains from many countries, including Argentina, grouped in this lineage. Lineage VI mainly comprised African human strains with P[4], P[6], and P[8] genotypes. Only a single bovine strain from Nigeria (NGRBg8; G8P[X]) was included in this lineage as well. Lineage VII consisted of only three strains, the first one (492SR, G8P[1]) infected a child in Paraguay [79], the second one (Y136, G8P[11]) was identified in a diarrheic calf from Brazil [39], and the third one (1115, G8P[1]) was obtained from a Peruvian alpaca [78]. Lineage VIII included human G8P[14] strains from Greece (057, 1100, 2885, and 2887) and Egypt (EGY1850) (Figure 3A).

The mean intra-lineage distance values ranged from 0.4 to 7.3%, whereas the mean inter-lineage distance values ranged from 12.4 to 17.8% (Figure 3B). According to the NJ tree topology (Figure 3A) and the pairwise distance frequency graph (Figure 3C), the cutoff value that best discriminates G8 lineages was determined at 10% genetic distance (90% nt identity) (see dotted line (Figure 3C)). The strains showing distance values lower than 10% frequently belonged to the same lineage, and strains with higher distance values frequently belonged to different lineages (Figure 3C).

### 3.6. Phylogenetic Analyses of VP7 and VP8* among G6 and G10 Strains

Twenty-two RVA strains from diarrheic beef and dairy calves of our study were selected for phylogenetic analyses. The G/P genotypes and G6 lineages [G6(III) or G6(IV)] determined by both SnM RT-PCR assays were in complete agreement with the results obtained by the Rotavirus A Genotype Determination Tool (ViPR) and the phylogenetic analysis of VP7, respectively.

The ML trees of VP7-G6 and VP8*-P[5] sequences showed that 10 strains from diarrheic beef calves segregated in two subclusters either within the G6(IV) (Appendix A) and P[5](I) (nine strains) lineages (Appendix A). One subcluster mainly grouped G6(IV)P[5](I) strains from Uruguay (Castells et al., 2020) together with five strains from the subregion “Cuenca del Salado”. The other subcluster grouped G6(IV)P[5](I) strains reported previously in Argentina [41] together with the five strains from the subregion “Depresion de Laprida”; these strains were phylogenetically very closely related (Appendix A). Of note, within the G6(IV) lineage, the strain 5595DR_D_BA obtained from a diarrheic dairy calf was closely related to the North American bovine strain B641 (Appendix A). The ML tree of VP7-G6 sequences also showed that six strains from diarrheic dairy calves were segregated in different branches of the G6(III) lineage. These branches included bovine P[11] strains from Argentina [41] and Uruguay [14] (Appendix A). Likewise, in the ML tree of VP8*-P[11] sequences, most of these strains clustered within the P[11](VI) lineage in different branches (Appendix A).

The ML analysis of VP7-G10 sequences revealed that three strains from diarrheic dairy calves clustered in the G10(VI) lineage. Two strains showed a very close phylogenetic relationship with bovine G10P[5] strains from Argentina [41], whereas strain 3DR_D_BA displayed a rather close relationship with bovine G10P[11] strains from Uruguay [14] (Appendix A). Regarding the phylogenetic analysis of P[11] sequences, two strains segregated in the P[11](VI) lineage, whereas strain 3DR_D_BA showed a rather close relationship with Uruguayan strains, all of which may represent a new lineage. The only VP8*-P[11] sequence obtained from a diarrheic beef calf (strain 11LC_B _BA; G6(IV)/G10P[11]) clustered in the P[11](VI) lineage and was closely related to strain 27412VA_ D_BA (G6(III)P[11]), which was collected from a diarrheic dairy calf (Appendix A).

### 3.7. Phylogeny of G6 RVA Strains According to Production Systems and Farm Location

To evaluate the phylogenetic relationship of G6(IV)P[5] and G6(III)P[11] strains according to the production system and the location of the farm within production regions, a selection of RVA strains from this and previous studies [41,44] was analyzed using the VP7 and VP8* sequences available in GenBank. Only strains from Buenos Aires province were included in the analysis. The clustering patterns are shown in NJ trees in Figure 4.

For the VP7 sequences, the G6(IV) lineage strains from the subregion “Depresion de Laprida” grouped within subcluster-1 (99% nt identity). Similarly, for the VP8*-P[5] sequences, these strains grouped together within the subcluster-2 (99% nt identity). In contrast, for the VP7 sequences, most of the G6(IV) lineage strains from the subregion “Cuenca del Salado” grouped within subcluster-2 (99% nt identity) and subcluster-3 (98% nt identity). Also, for the VP8*-P[5] sequences, most of these strains grouped within subcluster-1 (99% nt identity). Regarding the VP7 and VP8* sequences of RVA strains belonging to the G6(III) lineage and P[11] genotype, respectively, the NJ trees showed no clustering pattern according to the farm location (Figure 4A,B). Notably, according to our records, almost all G6(IV)P[5] strains were derived from beef calves, whereas almost all G6(III)P[11] strains were derived from dairy calves; reflecting that beef and dairy farms are located in different production regions of Buenos Aires province (Figure 4C).

### 3.8. Phylodynamic and Phylogeographic Analyses of G6(III) and G6(IV) Lineages

In American countries, and particularly in Argentina, the evolution and possible origin of bovine RVA strains belonging to G6(III) and G6(IV) lineages remain unknown. To answer these evolutionary questions, we prepared the RVA-G6 dataset that comprised 198 VP7 gene sequences of G6(III) (36%, 71/198) and G6(IV) (64%, 127/198) lineages. The RVA-G6(III) dataset included 71 VP7 sequences of RVA strains from three South American countries, namely Argentina (ARG) (45%, 32/71), Brazil (BRA) (34%, 24/71), and Uruguay (URY) (21%, 15/71). Conversely, the RVA-G6(IV) dataset included 127 VP7 sequences of RVA strains from three North American and four South American countries, namely ARG (34.6%, 44/127), BRA (46.4%, 59/127), Canada (CAN) (1.6%, 2/127), Mexico (MEX) (1.6%, 2/127), the United States of America (USA) (1.6%, 2/127), URY (13.4%, 17/127), and Venezuela (VEN) (0.8%, 1/127). The G6(III) lineage strains were restricted to South America, whereas the G6(IV) lineage strains were distributed across America.

The evolutionary rate for all of the G6 strains together was estimated at 1.24 × 10^−3^ nt substitutions/site/year (95% HDP, 9.97 × 10^−4^–1.52 × 10^−3^). This value was similar to those obtained for the G6(III) and G6(IV) lineage strains separately (Table 4).

According to the spatio-temporal reconstruction of all G6 strains together, the most probable ancestral location for the G6(III) lineage strains was ARG in the 1980s (95% HDP, 1963–1989), while for the G6(IV) lineage strains this was the USA in the 1940s (95% HDP, 1919–1959) (Table 4; Figure 5A). Similar dates were obtained for the RVA-G6(III) and RVA-G6(IV) datasets analyzed separately (Appendix A). The spread processes of bovine G6 RVA strains were inferred. The highest migration rates (BF > 1000) for all G6 strains together were from ARG to BRA and ARG to URY (Figure 5B). This last route of spread was also decisive (BF >1000) either for G6(III) or G6(IV) lineage strains (Appendix A). Another route of spread with strong support (100 < BF < 1000) for all G6 strains was from the USA to ARG (Figure 5B). Routes of diffusion with low support (3 < BF < 100) for the G6(IV) lineage strains were from CAN to ARG and USA to BRA. Other low-supported pathways were observed between BRA and VEN or CAN and MEX (Appendix A).

## 4. Discussions

NCD generates important economic losses due to increased mortality, treatment costs, and poor growth [16]. Epidemiological studies of enteric pathogens in many countries have demonstrated that RVA and *Cryptosporidium* spp. are the most frequent causes of NCD [8,9,10,12,13,80]. Hence, ongoing epidemiological studies are of great importance to assess the impact of RVA infections on livestock health. In this study, RVA was detected in at least one calf in ˃80% of beef and dairy farms during NCD outbreaks. Similarly, in the UK and Spain, calves shedding RVA were found in 58% to 93% of beef and dairy farms [8,13,81]. The broad distribution of this virus is due to its environmental resistance, which allows for long persistence in calf feces (up to 9 months) [82], the short period of incubation, and the large amount of virus shed by diarrheic calves for several days, even after clinical recovery [83].

Notably, few studies have compared the occurrence of RVA infections in diarrheic and non-diarrheic beef and dairy calves [8,12,13,17]. As expected, a significant association between RVA infection and diarrhea was observed in both beef and dairy calves. The frequencies of RVA infections in diarrheic calves (25.6%) agrees with previous studies conducted in Australia (31%) [22], Brazil (26.2%) [25], and Sweden (22.6%) [17]. Despite using the same Ag-ELISA, others in Argentina found a higher frequency of RVA infections (42%) [44]. Similarly, using Ag-ELISA or immunochromatography (IC) as detection methods, several studies have recorded higher frequencies in Denmark (46%) [52], France (49%) [84], Italy (37%) [85], Spain (51%) [13], Switzerland (58.7%) [9], and the UK (42% and 50.3%) [8,81]. Moreover, using real-time PCR and RT-PCR as detection methods, even higher rates have been observed in Australia (80%) [10], France (60%) [23], and Uruguay (57%) [14]. Differences among studies depend on many factors including the duration of the study, type of herd (beef or dairy), region, methods for selecting samples and herds (random or pre-selected), sample size, and case definition [84]. Also, detection methods based on the amplification of the viral genome have more sensitivity than Ag-based methods [10].

In this study, the epidemiology of RVA showed a different trend in beef and dairy farms. The frequency of RVA infections in both diarrheic and non-diarrheic calves was higher in beef farms. This finding supports previous studies conducted in Argentina and Brazil (beef calves, 23% and 46% vs. dairy calves, 16% and 24%) [80,86]. According to these data, NCD associated with RVA infections has a greater impact on beef cattle [82]. In beef herds, bull breading usually lasts 90–120 days, which results in many calvings during a short period. Thus, virus transmission between calves of similar age is enhanced by their gregarious behavior (crowding) and direct contact with feces and contaminated udders [82,86]. Also, at the beginning of the calving season, calves are usually infected subclinically and become biological amplifiers [87]. Conversely, in the dairy farms of our region, calves are frequently reared in individual stakes or hutches until weaning at two months of age. This practice may reduce virus transmission by direct contact. Subclinical RVA infections have been seldomly investigated [14,88]. In our study, RVA was detected in non-diarrheic beef (10%) and dairy (<5%) calves. Virus shedding during the incubation period or after clinical recovery can occur in infected calves [89], but infections can remain subclinical when high titers of specific passive maternal antibodies are present in the intestinal lumen [83].

The surveillance of G/P types among bovine RVA field strains is important because it allows for the discovery of new or emerging strains [20,90], provides insights into the epidemiology of animal infections [18] which may represent interspecies transmission events [77,90,91], and encourages the development and evaluation of vaccines [16,19,25,92]. Our results are in line with previous data available in five continents where G6 (56.7%), P[5] (25.9%), and P[11] (21.5%) are the most frequent genotypes infecting cattle [16]. However, we observed consistent differences in the frequency and diversity of G/P genotypes according to the production system. The G6(IV)P[5] combination was the most frequent in beef calves (≈90%), while the G6(III)P[11] combination and mixed genotypes prevailed in dairy calves (≈60%). These findings agree with a previous study in Argentina [11]. However, we observed that mixed genotypes and mixed RVA outbreaks were very uncommon in beef calves and farms, respectively. In contrast, mixed RVA outbreaks were very frequent in dairy farms, as reported previously [44]. Remarkably, only three studies report a different distribution of G/P types according to the production system. In beef herds from Sweden and the USA, G6 (67% and 100%) was more frequent than G10 (47.5%). Conversely, in dairy herds, G10 (17.5% and 54%) and G6 + G10 (10%) prevailed over G6 (32%) [17,93]. In Brazil, G6P[5] (65.5%) was most frequent in beef herds, while G10P[11] (38.4%) and G6P[11] (30.8%) prevailed in dairy herds [25]. Future studies should consider that production systems and cattle breeds, among other factors, may bias the fitness of different RVA genotypes.

So far, in Argentina and other countries, it is unknown which factors govern the differential distribution of G/P genotypes in beef and dairy farms. Probably, livestock management practices, breeds, and the unusual exchange of cattle between beef and dairy farms are involved. Beef herds are managed extensively with a calving season during winter. Thus, highly virulent strains should somehow persist in the environment and infect susceptible calves the next year or circulate subclinically in adult cattle for long periods [11,44]. The G6(IV)P[5] strains appear to have these features [11], and the lack of susceptible calves during a large part of the year could explain the low diversity of G/P genotypes and lineages observed in beef farms. In contrast, in dairy farms calves are usually born all year-round [44], and the failure of passive transfer of colostral antibodies is frequent [94]. This situation continuously offers susceptible calves, which enhances the opportunity for infections by less virulent strains [11] or co-infections by different strains. Consequently, genome reassortments are more likely to occur in dairy calves, which together with point mutations, increase the genetic diversity of RVA strains [16,91].

Surveillance of G/P types in circulating RVA strains is important for vaccine development [23] and vaccination programs [21,85]. Of note, G/P genotyping addresses the serotype specificity of these strains with respect to vaccines used on farms [21,85,92,95]. Vaccines available in Argentina include a G6(IV)P[5] strain (UK), but few of them also contain a G10P[11] strain (B223). In our study, 86% of beef and 55.5% of dairy farms performed systematic vaccination of pregnant dams. Despite this practice, G6(IV)P[5] was the most prevalent combination among RVA strains circulating in beef farms. This finding is consistent with available data in France, where G6(IV)P[5] was prevalent either in vaccinated (76.2%) and non-vaccinated (85.7%) herds [23]. Thus, vaccine-induced selective pressure is unlikely to occur in the beef herds of our region. However, the currently available vaccines may not offer protection against all circulating G/P types. Firstly, although we observed low circulation of G10P[11] strains, only cattle vaccinated with the B223 strain (G10P[11]) can develop neutralizing antibody responses against this strain if previous natural exposure has not occurred [96]. In cattle and guinea pigs, antibodies induced by the vaccine strain UK (G6(IV)P[5]) are not protective against strain B223 (G10P[11]) [96]. Secondly, the increase in emergent G/P combinations such as G8P[11] [this study, 95], G15P[11] [76,97], G5P[7] [42], and G24P[33] [14], or changes observed within the G6 genotype (G6P[11] strains) [this study, 21, 44, 92] could have an impact on vaccination programs, as the current vaccines may fail to protect [21,92].

During the study period, another interesting finding was the temporal variation of G/P genotypes in the dairy farm DR (Table 1). A single genotype combination (G6(III)P[11], G6(IV)P[5], and G10P[11]) was detected per year of sampling, suggesting yearly reintroductions. Circulating bovine RVA strains can show temporal fluctuations in their G/P genotypes [21,24]. In Ireland, G6 and P[5] displayed a decrease from 2004 to 2005 and then increased in 2009, while G10 and P[11] were coupled with steady increases from 2004 to 2009 [21]. According to our study, it is more likely to observe genotype fluctuations in dairy farms, and commercial bivalent vaccines (G6P[5] and G10P[11] strains) are expected to confer more protection than monovalent [21].

G8 RVA strains are distributed worldwide and infect a broad range of animal species including cattle [16,29,38], goats [54], and South American camelids (alpaca, guanaco, and vicugna) [53,76,78]. Despite this host diversity, the G8 genotype (3.5%) was by far less frequent than G6 (56.7%) and G10 (20.6%) genotypes in bovine RVA strains circulating in twenty-four countries and five continents [16]. Thus, bovine G8 strains are uncommon [39] and our results confirm this observation. Only a handful of bovine G8 strains have been genetically characterized for all of the 11 gene segments [29,33,38,39]. Likewise, G8P[1] and G8P[14] strains have been sporadically detected in ≤5-year-old children with or without gastroenteritis [31,32,79,98,99]. These strains possessed genotype constellations designated as human bovine-like [100] and most likely represent interspecies transmission from animals (e.g., ruminants) to humans [32,33,79,98]. Moreover, reassortant G8 strains could arise in animal hosts before transmission to humans [98] or in humans after transmission from animals (zoonotic genes) [101]. Human-bovine reassortant G8P[8] strains emerged as a cause of gastroenteritis in a human population in Central Europe [36]. Therefore, genomic data for animal G8 strains are necessary to understand their ecology, epidemiology, and evolution [40].

The strain 4385VT_D_BA possessed a backbone genotype constellation (non-G/P genotypes) that was shared with other RVA strains (I2-R2/R5-C2-M2-A3/A11/A13-N2-T6-E2/E12-H3) found in bovine, bovine-like, and artiodactyl bovine-like G8 strains infecting diverse animal species of the Bovidae (cattle, sheep, and goat), Cervidae (roe deer, among others), and Camelidae (guanaco and vicugna) families, as well as humans [27,28,38,39,53,54,76,101,102,103]. Of note, the R5 genotype is particularly interesting because R5-carrying strains have been described so far in camelids and cattle from South America [27,39,54,78]. Exhaustive phylogenetic analyses are necessary to unravel the most likely origin of G8 strains and assess their genetic relatedness to other strains [31,32].

Up to date, a single bovine G8P[11] strain (Y136) has been sequenced for all of the 11 gene segments [39]. However, only the VP7 gene was phylogenetically analyzed. The evolutionary analysis of strain 4385VT_D_BA showed that the VP3, VP4, VP6, and NSP1-NSP5 genes were very, or rather, closely related to the respective genes of RVA strains detected in calves from Argentina [76,77], Brazil [39], and Uruguay [14]. However, the VP7 and VP1 genes showed a very close relationship with the respective genes of the caprine bovine-like strain 0040 (G8P[1]), which was detected in a goat kid from a dairy farm in Argentina [54]. Also, the VP2 gene was rather closely related to that of strains 562 (G8P[14]) and 1115 (G8P[1]), which were obtained from Peruvian alpacas [78], as suggested by high similarity at the nt and aa level. Taken together, genomic data suggest at least two hypotheses explaining the origin of strain 4385VT_D_BA. Firstly, it could be a typical bovine RVA strain because its genotype constellation resembles those found in bovine strains [33,38,39,76], eight genes are phylogenetically derived from bovine strains, and two genes (VP7 and VP1) are closely related to those of the caprine bovine-like strain 0040, which is suspected to be a direct interspecies transmission from cattle [54]. However, the VP2 gene was rather closely related to strains from alpacas, which are South American camelids that do not inhabit Argentina. Secondly, strain 4385VT_D _BA could represent a typical bovine strain with at least one gene (VP7) originating through a reassortment event, between bovine and artiodactyl bovine-like strains after interspecies transmission (e.g., cattle, goats, and guanacos) [54,76,78]. This finding is not surprising because the evolutionary analysis of the E12-NSP4 genotype revealed that the guanaco strain Rio_Negro (G8P[1]) was ancestral with respect to all RVA strains from Argentina, Uruguay, and Paraguay. However, regarding VP1 and VP2 genes, more sequences from South American bovine RVA strains should be generated to obtain conclusive data.

At present, the intragenotype diversity of G8 is represented by six lineages [33,36,38,40], but a new lineage, tentatively designated G8(VII), has been proposed previously and comprises human (492SR, G8P[1]) and bovine (Y136, G8P[11]) strains from Paraguay and Brazil, respectively [39]. Additionally, phylogenetic studies of human G8 strains request extensive sequence analysis of the VP7 gene in order to establish the lineages correctly [40]. In this paper, we conducted a comprehensive genetic analysis of the G8 genotype, and eight lineages [G8(I) to G8(VIII)] were observed in the phylogram. In the pairwise distance frequency graph, the genetic distance values within and between lineages showed minor overlapping. However, it should be noted that previous studies and the current one have assigned certain human P[4] and P[14] strains (AU109, BP1062, and AS970) within the G8(V) lineage [33,38,39,40]. The high genetic distance values (10–12%) observed between these and other G8(V) lineage strains explain this limited overlap. Future studies should evaluate these strains as a possible new G8 lineage. Interestingly, after our analysis, the G8(VII) lineage included an alpaca strain (1115, G8P[1]) from Peru, which was reported recently [78]. Moreover, the G8(VIII) lineage proposed in the current study mainly grouped human G8P[14] strains from Greece. As expected, the collection of Argentinean G8 strains from artiodactyl species [53,54,76,78], which also included the bovine study strain, grouped together in the same branch of the G8(II) lineage, supporting a common origin for VP7.

Epidemiologically, few studies have evaluated in detail the occurrence of particular lineages of G6, G10, P[5], and P[11] genotypes. In Brazil [25], France [23], Ireland [21], the Netherlands [43], and Spain [13], G6(IV) (40–100% of strains) and G6(III) (12–20% of strains) are the most prevalent intragenotype lineages among bovine strains. In this study, the ML phylogenetic analysis of the VP7-G6 sequences revealed a particular pattern. The G6(IV) lineage comprised all P[5] strains from beef calves and only two from dairy ones, whereas the G6(III) lineage included all P[11] strains from dairy calves. This segregation according to the P genotype and production system (beef or dairy) agrees with previous studies conducted in Argentina [41] and Brazil [25]. It is unknown which factors cause this observation, but others also report a genetic linkage between G6(IV) and P[5], or G6(III) and P[11] [14,23,25,42]. The VP4 protein of the spike has many functions including binding and fusion to cell membrane, infectivity, virulence, and neutralization (antigen) [7]. Therefore, these features and host factors (i.e., breed of calves) may explain the restriction of P types according to the G6 lineage and production system.

Interestingly, the G6(IV) strains from the subregion “Cuenca del Salado” and Uruguay [14] were rather closely related, and this finding has not been observed before in South America [14,25,41,92,104]. Probably, this genetic relationship could be due to the geographic proximities between this region and Uruguay. Interestingly, the G6(IV) strain 5595DR_D_BA, which was obtained from a diarrheic dairy calf, was phylogenetically closely related to the North American bovine strain B641. Up to date, none of the South American strains displayed this phylogenetic feature, at least for VP7 [25,41,44,92,104]. Regarding the G10 genotype, 10 lineages (I-X) are recognized [105]. The G10 strains reported herein clustered within the G10(VI) lineage together with Argentinean [41] and Uruguayan strains [14]. In contrast, the Brazilian bovine G10 strains clustered apart within G10(III) and G10(IV) lineages [25]. Concerning the P[5] and P[11] genotypes, most of the strains characterized in this study clustered in the P[5](I) and P[11](VI) lineages. Of note, due to the increasing number of P[11] sequences, their lineage classification might have to be updated as three strains from this study could not be included in the previously proposed six lineages (I-VI) [25,41]. In this regard, the Uruguayan P[11] strains [14] together with strain 3DR_D_BA (G10P[11]) may represent a new lineage.

Although phylogenetic studies of circulating bovine RVA strains have been performed [13,14,20,22,23,25,42,43,90], none of them assessed the linkage of RVA strains according to the production system (beef or dairy) and the location of farms in production regions. The VP7 and VP8* sequences of the highly prevalent G6(IV)P[5] and G6(III)P[11] strains were analyzed; the sequences obtained in this and previous studies were included [41,44]. Both genes were useful to assess the phylogenetic relationship of G6(IV)P[5] strains according to the production system and farm location. In the future, this epidemiological approach should be considered in the phylogenetic studies of RVA strains infecting domestic animals.

Currently, the G6(III) and G6(IV) lineage strains are highly prevalent in calves with diarrhea [13,21,23,25,41,43]. The first bovine RVA strain, designated NCDV-Lincoln (G6P[1]), was described in the USA in 1969 [45] and belonged to the G6(IV) lineage [44]. Since its first description, phylodynamic and phylogeographic studies of bovine G6 strains seem not to have been conducted. These studies are important to gain knowledge about the evolution of pathogens and their possible geographical origins [106]. Although many factors influence evolutionary rates [106], the values obtained for G6 strains together (1.24 × 10^−3^ s/s/y), or for G6(III) (1.17 × 10^−3^ s/s/y) and G6(IV) (1.26 × 10^−3^ s/s/y) lineages separately, were similar and within the range (1 × 10^−2^–1 × 10^−5^ s/s/y) reported for RNA viruses [107,108]. Moreover, these values are consistent with estimations obtained for human and porcine RVA strains belonging to the lineage III of G9 (1.87 × 10^−3^ s/s/y) and G12 (1.66 × 10^−3^ s/s/y) genotypes [109]. Also, the evolution of VP7 among human G2P[4] (1.14 × 10^−3^ s/s/y) and G4P[8] (1.92 × 10^−3^ s/s/y) strains are in agreement with our findings [106].

The spatio-temporal Bayesian inference of American bovine G6 RVA strains was performed using three VP7 gene sequence datasets. The RVA-G6 dataset, which included G6(III) and G6(IV) lineage strains together, showed that the diversification of bovine G6 RVA in the American countries would have started in the mid-1850s in the USA, almost a century before its first description as a cause of NCD in that country [45]. As suggested by the phylodynamic and phylogeographic data, G6 RVA would have spread from North to South America and expanded its genetic diversity in South American countries. However, the currently available VP7 sequences from North American strains are scarce and may bias our interpretations; therefore, further sequences are needed.

The ancestral location of the G6(IV) lineage would be the USA around 1940, but two large branches in the MCCT started to evolve in Brazil and then in Argentina. This finding should be interpreted with caution because, as discussed previously, the VP7 sequences of dated strains from the USA represented only 1.6%, whereas Brazilian and Argentinean sequences comprised 47.2% and 34.6%, respectively. Introductions of RVA in Brazil from other countries in addition to the USA should not be discarded; indeed, our analyses suggest virus introductions in Brazil from Argentina and Venezuela. The age and origin of the nearest common ancestor of the G6(III) lineage were around 1980 in Argentina. This finding is expected because this lineage was first reported in Argentina [44] and thereafter in Brazil [110] and Uruguay [14]. As indicated by phylogeographic data, the route of spread from Argentina to its neighboring countries seems plausible. In agreement with this finding, Argentina is suggested as the most probable origin and route of introduction for BCoV in Uruguay [111].

## 5. Conclusions

RVA infections are still a frequent cause of diarrhea in neonatal calves and can have a greater health impact on beef farms. The G/P genotypes, genetic diversity, and phylogeny of circulating bovine RVA strains can differ according to the production system and the location of farms. Thus, future studies should evaluate which factors may bias the fitness of RVA field strains in beef and dairy calves and farms. Also, our research provides further novel knowledge about the evolutionary origin of the G8 strains infecting different animal species. Finally, in American countries, the phylodynamic/phylogeographic data generated in this study support how bovine G6(IV) strains emerged first in the USA, nearly a century before its first report in calves from the same country. Conversely, bovine G6(III) strains emerged later in Argentina and were introduced in Uruguay from this neighboring country.

## Figures and Tables

**Figure 1 viruses-15-02115-f001:**
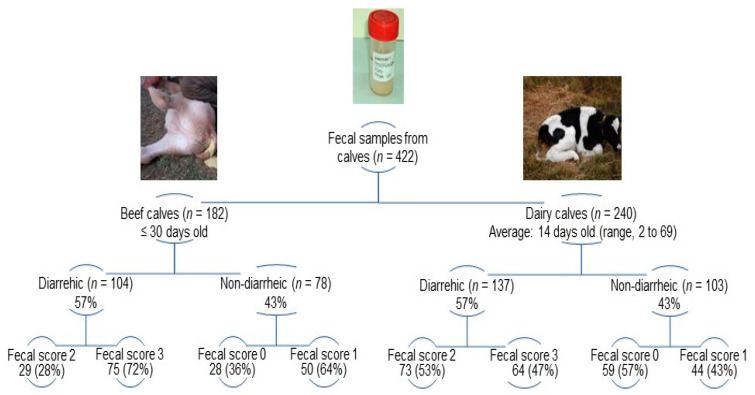
Distribution of fecal scores among sampled calves according to the production system (beef and dairy farms) and clinical condition (diarrheic and non-diarrheic).

**Figure 2 viruses-15-02115-f002:**
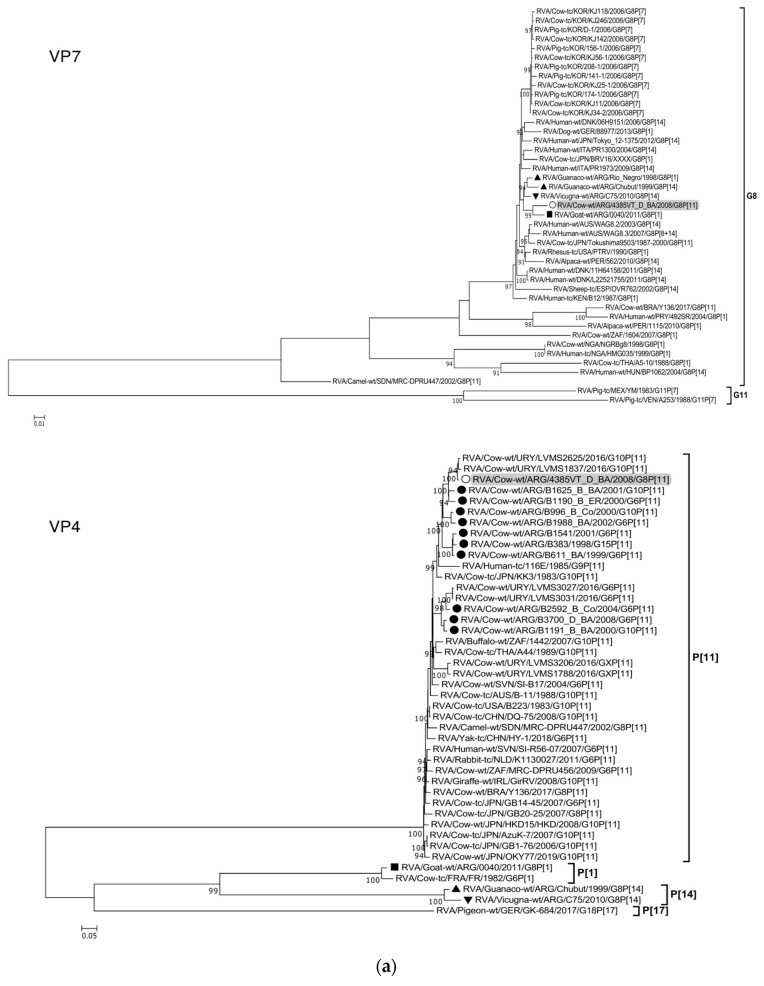
(**a**–**f**). Maximum likelihood phylogenetic trees of VP7, VP4, VP6, VP1, VP2, VP3, NSP1, NSP2, NSP3, NSP4, and NSP5 genes. The best fitting models of nucleotide substitution were: GTR + F + G4 (VP7, VP1, VP2, NSP2, NSP3), GTR + F + I + G4 (VP4, VP6, VP3, NSP1), TPM2 + F + G4 (NSP4), and TIM2 + F + I + G4 (NSP5). The analysis included similar sequences retrieved from GenBank as well as those from G8 and/or P[11] RVA strains (Table 3). Trees were rooted using genotypes other than those of strain 4385VT_D_BA, as follows: G11 (VP7); P[1], P[14], and P[17] (VP4); I1, I3, and I10 (VP6); R2 (VP1); C3 and C5 (VP2); M3 (VP3); A3, A11, and A18 (NSP1); N1 (NSP2); T1, T7, and T9 (NSP3); E1, E2, E3, and E5 (NSP4); H1 and H5 (NSP5). Ultrafast bootstrap values (10,000 replicates) ≥ 90% are shown as branch nodes. The bovine strain 4385VT_D_BA (white circle) is highlighted using a grey shadow. Other Argentinean strains from cattle (black circle), goat (black square), guanaco (black triangle), horse (black diamond), and vicugna (black inverted triangle) are indicated. Scale bars indicate nucleotide substitutions per site.

**Figure 3 viruses-15-02115-f003:**
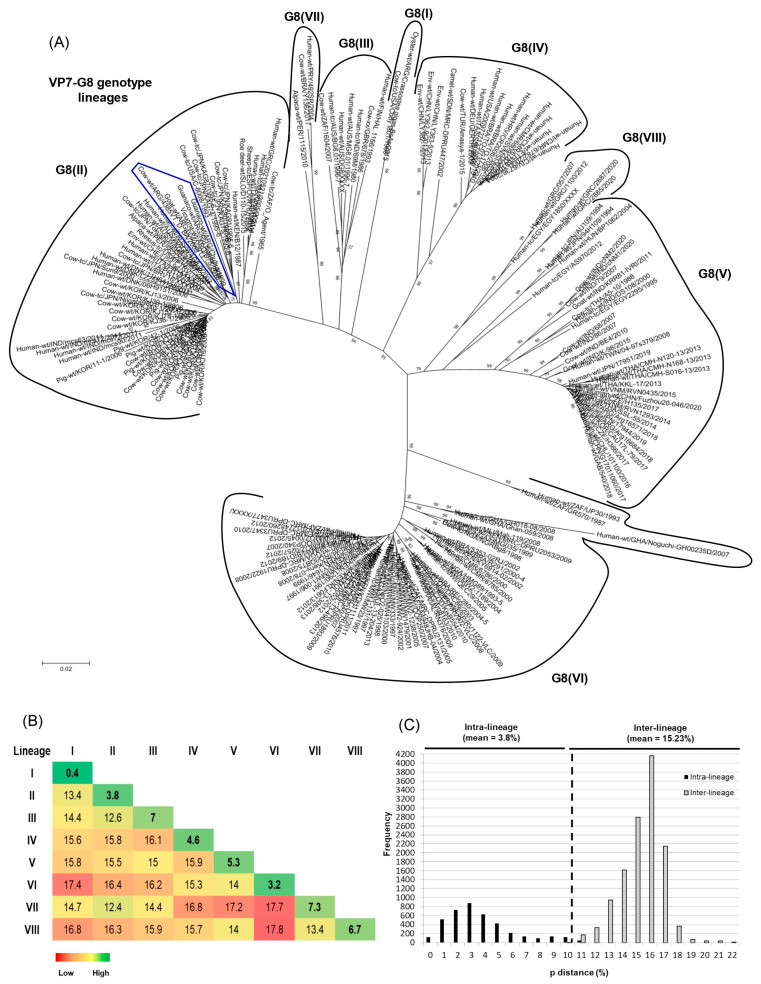
Phylogenetic analyses of G8 RVA strains. (**A**) Phylogenetic tree of the G8 genotype. The phylogram was reconstructed using the Kimura 2-parameter substitution model and the neighbor-joining method. For better visualization, lineages are surrounded by black lines and strain names are abbreviated. The G8 linages are designated according to previous studies [38,40] using Roman numbers. Argentinean G8 strains from this and previous studies are highlighted with an irregular blue pentagon. (**B**) Comparison of genetic distances within (intra-lineage) and between (inter-lineage) the G8 lineages. Genetic distance values, expressed as percentages, were calculated using the pairwise distance algorithm and the p-distance model (MEGA 7.0). (**C**) Frequency graph of pairwise genetic distances. The percentage values of genetic distances are indicated in the abscissa (x-axis) and the frequencies of these values are indicated in the ordinate (y-axis). The cutoff value to best discriminate lineages is shown by a dotted line.

**Figure 4 viruses-15-02115-f004:**
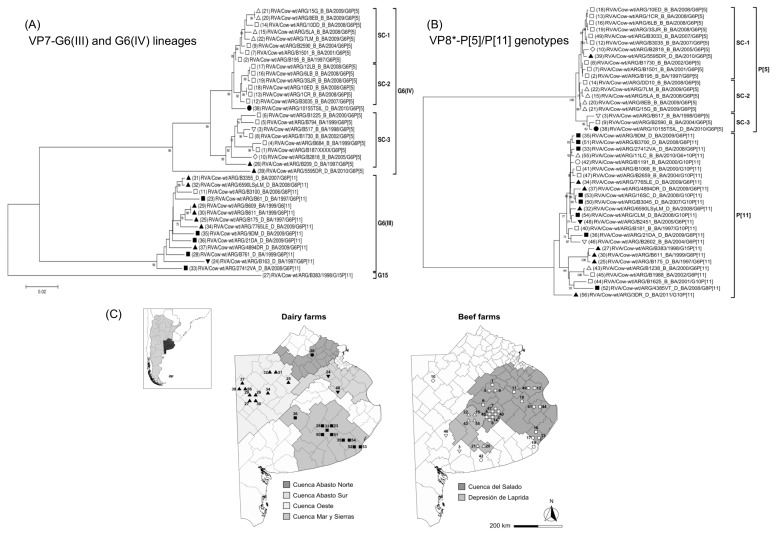
Phylogenetic trees of bovine RVA strains from Buenos Aires province, Argentina. (**A**) VP7 sequences of RVA strains corresponding to G6(III) and G6(IV) lineages. (**B**) VP8* sequences of RVA strains with P[5] and P[11] genotypes. Phylograms were reconstructed using the Kimura 2-parameter substitution model and the neighbor-joining method (MEGA 7.0 software). The phylogram of VP7-G6 lineages was rooted using strain RVA/Cow-wt/ARG/B383/1998/G15P[11]. The analysis only included similar sequences and with available information. Each strain is represented with a number and symbol; white symbols correspond to beef farms, whereas black symbols correspond to dairy farms, as follows: □ “Cuenca del Salado”; △ “Depresion de Laprida”; ■ “Mar y Sierras”; ▲ “Oeste”; ● “Abasto Norte”; ▼ “Abasto Sur”. Bootstrap values (1000 replicates) >70% are shown as branch nodes. The name of the strain, type of production system, and province of collection are indicated. Abbreviations: D/B, Dairy or Beef; BA, Buenos Aires. (**C**) The map of Buenos Aires province shows RVA strains according to the production system (dairy or beef) and the location of farms in districts and production regions; each strain is represented with numbers and symbols as indicated previously. The location of the farms does not represent GPS coordinates.

**Figure 5 viruses-15-02115-f005:**
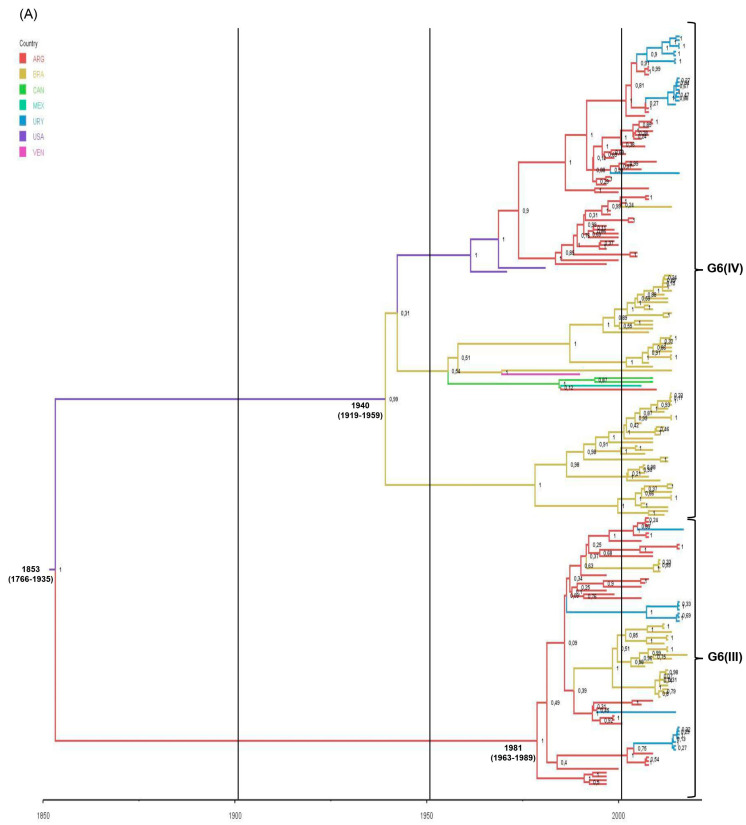
Phylodynamic and phylogeographic reconstruction of bovine G6 RVA strains in American countries. (**A**) Maximum clade credibility tree (MCCT) of G6 lineage III [G6(III)] and G6 lineage IV [G6(IV)] strains together. The color of the branches represents the most probable country of origin. Posterior probability values of main clades are shown. Time of the most recent common ancestor (TMRCA) and the 95% highest posterior density (95% HPD) intervals are indicated below the tree nodes. Country abbreviations using the three-letter code (alpha-3) (ISO 3166): ARG, Argentina; BRA, Brazil; CAN, Canada; MEX, Mexico; URY, Uruguay; USA, the United States of America; VEN, Venezuela. (**B**) Inferred spread pattern of G6(III) and G6(IV) lineage strains in American countries. Line color represents the relative strength of connection between countries according to the Bayes Factor (BF) test: red arrows, decisive support with BF > 1000; blue arrows, strong support with 100 < BF < 1000; and green arrows, supported rates with 3 < BF <100.

**Table 1 viruses-15-02115-t001:** Genotypes and G6 lineages detected in RVA strains infecting beef and dairy calves.

Genotypes and G/P Combinations	Beef (*n* = 33)	Dairy (*n* = 34)		Dairy Farm DR *		
				2009	2010	2011
	No. (%)		*p*-Value _FDR_ ^&^	No.		
G6(III)	0 (0.0)	15 (44.1)	<0.0001			
G6(IV)	30 (90.9)	4 (11.8)	<0.0001			
G8	0 (0.0)	1 (2.9)	0.1734			
G10	0 (0.0)	3 (8.8)	0.0016			
P[5]	32 (97.0)	7 (20.6)	<0.0001			
P[11]	1 (3.0)	24 (70.6)	<0.0001			
G6(III)P[5]	0 (0.0)	1 (2.9)	0.2033			
G6(IV)P[5]	30 (90.9)	3 (8.8)	<0.0001		7	
G6(III)P[11]	0 (0.0)	14 (41.2)	<0.0001	5		
G8P[11]	0 (0.0)	1 (2.9)	0.2049			
G10P[11]	0 (0.0)	3 (8.8)	0.0057			2
GXP[5]	2 (6.1)	2 (5.9)	0.9731			
GXP[11]	0 (0.0)	2 (5.9)	0.0639	2		
G6(IV) + G10P[11]	1 (3.0)	2 (5.9)	0.6841			
G6(IV) + G10P[5]	0 (0.0)	1 (2.9)	0.2033			
G6(III) + G10P[11]	0 (0.0)	2 (5.9)	0.0553			
G6(IV) + P[5 + 11]	0 (0.0)	1 (2.9)	0.2049			
G6(IV) + G10P[5 + 11]	0 (0.0)	2 (5.9)	0.0553			
Mixed genotypes (total)	1 (3.0)	8 (23.5)	0.0553			

* This farm was visited due to severe successive NCD outbreaks. The fecal samples collected in 2009 were included for statistical, genotype, and phylogenetic analyses, whereas the samples collected in 2010 and 2011 were included for genotype comparisons and phylogenetic analysis. ^&^ FDR: False Discovery Rate adjustment for multiplicity.

**Table 2 viruses-15-02115-t002:** Genome characterization of the unusual bovine G8P[11] RVA strain.

Genome Segment (Protein)	Sequenced Fragment ^†^ (bp)	Position *	ORF Coverage (%)	Genotype ^#^	Most Similar Strains after Nucleotide BLAST Search	Highest Identity (%) ^&^	Host of Origin	Country
1 (VP1)	3215	46-3260	98	R5	RVA/Cow-wt/ARG/B383/1998/G15P[11]	98	Cattle	Argentina
2 (VP2)	2643	17-2659	100	C2	RVA/Alpaca-wt/PER/1115/2010/G8P[1]	96	Alpaca	Peru
3 (VP3)	1518	50-1567	61	M2	RVA/Guanaco-wt/ARG/Chubut/1999/G8P[14]	91	Guanaco	Argentina
4 (VP4)	2142	163-2304	92	P[11]	RVA/Cow-wt/ARG/B1625_B_BA/2001/G10P[11]	95	Cattle	Argentina
5 (NSP1)	1427	83-1509	95	A13	RVA/Cow-wt/ARG/B383/1998/G15P[11]	94	Cattle	Argentina
6 (VP6)	1167	51-1217	98	I2	RVA/Cow-wt/URY/LVMS3206/2016/GXP[11]	93	Cattle	Uruguay
7 (VP7)	898	49-946	92	G8	RVA/Goat-wt/ARG/0040/2011/G8P[1]	97	Goat	Argentina
8 (NSP2)	954	47-1000	100	N2	RVA/Cow-wt/URY/LVMS3053/2016/G10P[X]	98	Cattle	Uruguay
9 (NSP3)	933	35-967	100	T6	RVA/Cow-wt/URY/LVMS2625/2016/G10P[11]	99	Cattle	Uruguay
10 (NSP4)	528	42-569	100	E12	RVA/Cow-wt/ARG/B3700/2008/G6P[5]	99	Cattle	Argentina
11 (NSP5)	597	22-618	100	H3	RVA/Cow-wt/ARG/B383/1998/G15P[11]	99	Cattle	Argentina

^†^ Sequences were deposited in GenBank under the accession numbers OR344101 (VP7), OR344102 (VP4), OR344103 (VP6), OR344104-OR344106 (VP1-VP3), and OR344107-OR344111 (NSP1-NSP5).* Nucleotide (nt) positions and ORF coverage (%) values were obtained according to the complete genomes of strains RVA/Guanaco-wt/ARG/Chubut/1999/G8P[14] (VP7) and RVA/Cow-wt/ARG/B383/ 1998/G15P[11] (VP4, VP6, VP1-VP3, and NSP1-NSP5) [76]. ^#^ Genotyping was conducted according to the guidelines of the RCWG [27] and using the Rotavirus A Genotype Determination tool, which was available in the ViPR database [57]. ^&^ The nt sequence identity (%) was estimated using the Nucleotide BLAST tool (https://blast.ncbi.nlm.nih.gov/Blast.cgi; accessed on 27 March 2023).

**Table 3 viruses-15-02115-t003:** Comparison of genotype constellations between the bovine G8P[11] RVA strain and other reference G8 and/or P[11] strains.

Strains	Genotype Constellation	VP7 (%) *	VP4 (%)	VP6 (%)	VP1 (%)	VP2 (%)	VP3 (%)	NSP1 (%)	NSP2 (%)	NSP3 (%)	NSP4 (%)	NSP5 (%)	Shared Genotypes
**RVA/Cow-wt/ARG/4385VT_D_BA/2008/G8P[11]**		**G8**	**P[11]**	**I2**	**R5**	**C2**	**M2**	**A13**	**N2**	**T6**	**E12**	**H3**	
RVA/Cow-wt/ARG/B383/1998/G15P[11]	Bovine	G15	**P[11]** **(94/94)**	I2 (84/97)	**R5 (98/100)**	C2 (93/99)	M2 (88/95)	**A13 (94/94)**	**N2 (95/98)**	T6 (88/96)	**E12 (97/99)**	**H3 (99/100)**	**10**
RVA/Guanaco-wt/ARG/Rio_Negro/1998/G8P[1]	Artiodactyl bovine-like	**G8** **(97/99)**	P[1]	I2 (91/99)	R5 (96/99)	C2 (85/98)	M2 (84/92)	A11	N2 (87/95)	T6 (89/97)	E12 (94/97)	H3 (93/94)	**9**
RVA/Goat-wt/ARG/0040/2011/G8P[1]	Artiodactyl bovine-like	**G8** **(97/98)**	P[1]	I2 (87/97)	R5 (98/99)	**C2 (96/100)**	M2 (88/94)	A3	**N2 (95/98)**	T6 (94/98)	E12 (96/99)	H3 (93/94)	**9**
RVA/Guanaco-wt/ARG/Chubut/1999/G8P[14]	Artiodactyl bovine-like	G8 (97/97)	P[14]	I2 (91/99)	R5 (93/99)	C2 (85/98)	**M2 (91/95)**	A3	N2 (91/97)	**T6 (98/99)**	**E12 (96/100)**	H3 (96/97)	**9**
RVA/Rhesus-tc/USA/PTRV/1990	Artiodactyl bovine-like	G8 (96/98)	P[1]	I2 (91/99)	R2	C2 (85/98)	M2 (85/92)	A3	N2 (94/99)	T6 (95/98)	E2	H3 (97/99)	**9**
RVA/Human-tc/KEN/B12/1987/G8P[1]	Human bovine-like	G8 (96/99)	P[1]	I2 (90/99)	R2	C2 (86/98)	M2 (84/92)	A3	N2 (88/96)	T6 (97/98)	E2	H3 (96/98)	**9**
RVA/Alpaca-wt/PER/562/2010/G8P[14]	Artiodactyl bovine-like	G8 (95/98)	P[14]	I2 (90/99)	R5 (91/98)	C2 (96/100)	M2 (80/88)	A17	N2 (96/97)	T6 (94/98)	E3	H3 (97/99)	**8**
RVA/Cow-wt/BRA/Y136/2017/G8P[11]	Bovine	G8 (87/94)	P[11] (71/71)	I2 (88/98)	R5 (89/99)	C2 (84/96)	M2 (89/93)	A3	N2 (87/98)	T9	E2	H3 (96/98)	**8**
RVA/Alpaca-wt/PER/1115/2010/G8P[1]	Artiodactyl bovine-like	G8 (88/93)	P[1]	**I2** **(91/100)**	R2	C2 (96/99)	M2 (84/92)	A17	N2 (92/98)	T6 (87/96)	E3	H3 (97/100)	**7**
RVA/Cow-wt/ZAF/1604/2007/G8P[1]	Bovine	G8 (87/98)	P[1]	I2 (91/99)	R2	C2 (85/98)	M2 (85/91)	A3	N2 (94/97)	T6 (89/97)	E2	H3 (95/97)	**7**
RVA/Cow–tc/NGA/NGRBg8/1998/G8P[1]	Bovine	G8 (85/96)	P[1]	I2 (87/99)	R2	C2 (85/98)	M2 (81/91)	A11	N2 (88/97)	T6 (94/98)	E2	H3 (95/98)	**7**
RVA/Cow-tc/THA/A5-10/1988/G8P[1]	Bovine	G8 (85/96)	P[1]	I2 (92/99)	R2	C2 (84/98)	M2 (82/91)	A11	N2 (87/96)	T6 (92/98)	E2	H3 (95/97)	**7**
RVA/Cow-tc/CHN/DQ-75/2008/G10P[11]	Bovine	G10	P[11] (72/72)	I2 (91/99)	R2	C2 (84/98)	M2 (82/90)	A3	N2 (94/98)	T6 (92/97)	E2	H3 (95/97)	**7**
RVA/Human–tc/NGA/HMG035/1999/G8P[1]	Human bovine-like	G8 (85/96)	P[1]	I2 (87/99)	R2	C2 (85/98)	M2 (81/91)	A11	N2 (88/97)	T6 (94/98)	E2	H3 (95/98)	**7**
RVA/Human-wt/PRY/492SR/2004/G8P[1]	Human bovine-like	G8 (87/94)	P[1]	I2 (87/97)	R2	C2 (85/98)	M1	--	N2 (95/96)	T6 (95/97)	E12 (96/99)	H3 (99/99)	**7**
RVA/Human-wt/ITA/PR1300/2004/G8P[14]	Human bovine-like	G8 (96/98)	P[14]	I2 (87/99)	R2	C2 (84/98)	M2 (82/90)	A3	N2 (88/97)	T6 (90/98)	E2	H3 (97/100)	**7**
RVA/Human-wt/HUN/BP1062/2004/G8P[14]	Human bovine-like	G8 (83/95)	P[14]	I2 (91/99)	R2	C2 (84/97)	M2 (81/91)	A11	N2 (87/97)	T6 (91/98)	E2	H3 (95/97)	**7**
RVA/Sheep-tc/ESP/OVR762/2002/G8P[14]	Artiodactyl bovine-like	G8 (94/98)	P[14]	I2 (90/99)	R2	C2 (84/98)	M2 (81/90)	A11	N2 (88/96)	T6 (91/98)	E2	H3 (97/98)	**7**
RVA/Vicugna-wt/ARG/C75/2010/G8P[14]	Artiodactyl bovine-like	**G8** **(97/99)**	P[14]	I2 (90/99)	R2	C2 (85/96)	**M2 (92/96)**	^--^	N2 (92/97)	T6 (97/98)	E3	--	**6**
RVA/Human-wt/GHA/GH018-08/2008/G8P[6]	Human bovine-like	G8 (85/96)	P[6]	I2 (91/99)	R2	C2 (85/98)	M2 (84/92)	A2	N2 (89/97)	T2	E2	H3 (97/99)	**6**
RVA/Dog-wt/GER/88977/2013/G8P[1]	Dog bovine-like	G8 (95/98)	P[1]	I2 (92/99)	-- ^&^	C2 (85/99)	--	A3	N2 (88/97)	T6 (90/98)	E2	H3 (97/99)	**6**

* Nucleotide (nt) and deduced amino acid (aa) sequence identity values (nt/aa), expressed as percentages, were estimated using the complete or partial ORF sequences of each gene segment. Distance matrixes of nt and aa sequences were obtained applying the p-distance algorithm in MEGA 7.0 [58]. Shared genotypes are highlighted in shaded grey, whereas high identity values are indicated in bold font. ^&^ Gene sequences not obtained in the original research.

**Table 4 viruses-15-02115-t004:** Evolutionary rate, time of the most recent common ancestor (TMRCA), and most probable American country of origin for bovine G6(III) and G6(IV) lineage strains analyzed in this study.

	**VP7 ORF Sequence Datasets**		
Parameter	RVA-G6	RVA-G6(III)	RVA-G6(IV)
n	198	71	127
Sampling time interval	1971–2018	1997–2018	1971–2016
Evolutionary rate (Mean s/s/y) ^&^ *	1.24 × 10^−3^ (9.97 × 10^−4^–1.52 × 10^−3^)	1.17 × 10^−3^ (6.49 × 10^−4^–1.72 × 10^−3^)	1.26 × 10^−3^ (9.44 × 10^−4^–1.58 × 10^−3^)
Mean TMRCA ^#^ * for G6 strains	1853 (1766–1935)	NA	NA
Mean TMRCA ^#^ * for G6(III) lineage	1981 (1963–1989)	1982 (1971–1992)	NA
Mean TMRCA ^#^ * for G6(IV) lineage	1940 (1919–1959)	NA	1945 (1926–1962)
Country ^†^ (probability)	USA (0.57) ARG (0.23)	ARG (0.99)	USA (0.55) BRA (0.29)

^&^ Substitutions/site/year. ^#^ Time of the most recent common ancestor (TMRCA). * The 95% highest posterior density (95% HPD) interval is shown within parentheses. NA: not applicable. ^†^ Country abbreviations using the three-letter code (alpha-3) (ISO 3166): ARG, Argentina; BRA, Brazil; USA, the United States of America.

## Data Availability

The nucleotide sequences were deposited in GenBank under the accession numbers OR253956-OR253973 (VP7-G6), OR253974-OR253976 (VP7-G10),OR269758-OR269768 (VP8*-P[5]), OR269769-OR269778 (VP8*-P[11]), OR344101 (VP7-G8), OR344102 (VP4-P[11]), OR344103 (VP6), OR344104-OR344106 (VP1-VP3), and OR344107-OR344111 (NSP1-NSP5). The larger aligned sequence datasets, which were designated RVA-G8(lineages) (184 VP7 sequences), RVA-G6 (198 VP7 sequences), RVA-G6(III) (71 VP7 sequences), and RVA-G6(IV) (127 VP7 sequences), are available with GenBank accession numbers in the Appendix A.

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
