# Peer review of "Molecular Epidemiology of Rotavirus A in Calves: Evolutionary Analysis of a Bovine G8P[11] Strain and Spatio-Temporal Dynamics of G6 Lineages in the Americas"

_viruses, 2023, doi:10.3390/v15102115_

Round 1
Reviewer 1 Report
Uriarte et al. identified RVA strains during the outbreaks in past decades. They used a good sample size to analyze the data and interpretation. They showed specific genotype prevalence for beef and calves due to unknown reasons which is beyond the scope of this study. Phenotypic variabilities among RNA viruses are not new but enteric viruses’ tropism still lacking in the field for example recently rotavirus was found to infect tuft cells, and norovirus infects salivary glands, and such biology is important in controlling the disease. The authors used convincing methods to interpret their findings, however, this is more of a computational study. The finding is important to the field especially the readers of virus evolution. I think including limitations of this study at the end will be useful for the readers.
Major comments:
1. Authors identified G6(IV)P[5] as a prevalent strain in beef (91%) and G6(III)P[11] in calves (41%), interestingly significant mixed strains were only identified in calves. However, RVA infection rate is higher in beef than in calves, suggesting that G6(IV)P[5] has less infectivity in calves. What are the major differences between beef and calves that can explain this ambiguity?
2. How this tropism differs, what are the known receptors that may potentially influence the infection rate?
3. Can authors make a pictorial representation of no. of animals, diarrhea incidence separated by scores, and their ages at the time of stool collection? This should go into main figures, not supplementary ones. This will be easy for readers to follow, but difficult to follow the number at different sections.
4. What is the meantime of diarrhea persistence in calves, and at which stage (eg. peak or decline stage) the stools were collected for analysis?
5. Line150: why vaccination of dams is limited not 100% vaccinated provided RVA can transmit through oral fecal rate? How do these beef and calves behave between vaccinated and no vaccine, incidence of diarrhea with score?
6. Do authors calculate the viral copy number in the different fecal scoring patterns?
7. List the primers and RT-PCR condition used in this study and include pictures of a few if not all the samples in supplementary figures. Here multiple bands are misleading (explain the different band size that is expected using the primer pairs).
8. Do authors find a correlation between ELISA and RT-PCR?
9. Authors should include the deposited or innovated pipelines used for computational analysis.
Minor comments
Line 1: Possesses is correct word instead of possess
Line 103: diversifications instead of diversification
Line 107: linkages instead of linkage
Line 643: frequencies instead of frequency
Line 655: method instead of methods
Line 734: strain not strains
Line 807: proximities instead of proximity
Author Response
"Please see the attachment."

Reviewer 2 Report
This is a large study on epidemiology and genotypes of group A rotaviruses in calves in Argentina and Uruguay. The study samples (total 422) were collected from a total of 32 farms between 2007-2010, and, apparently, have been analysed over a long period of time. The samples were extensively genotyped. Prevalent lineages were G6, either P[5] or P[11]. G8 formed a minority but represent a more novel discovery. The most significant contribution of this report may be the detailed genomic characterization and phylogenetic analyses of the bovine G8 strains.
The study is fine as such, the paper well written and it could be published as it is. However, the paper could be improved by adding clinical points. In the paper there is very little about the diarrhea in calves, just that more samples were collected from diarrheic than healthy calves. How significant was the diarrhea? The paper says that diarrhea was more of a problem in beef than dairy calves, because of crowding in the former. Crowding as a factor in transmission of rotavirus in calves has been reported before, and it could be mentioned.
What is missing is the relevance of genotyping of bovine rotaviruses for rotavirus vaccination in calves. Is this information important for vaccine development and use? I do not believe there really are serotype-specific rotavirus vaccines for cows and even more importantly they may not be needed. The authors should comment on the current situation of rotavirus vaccination in cattle in relation to bovine rotavirus genotypes.
Author Response
"Please see the attachment."

Reviewer 3 Report
The authors described the evolutionary origin of a bovine G8P[11] strain, which had zoonotic potential for direct transmission. They also demonstrated that bovine G6 strains from American countries would have originated in the USA nearly a century before its first description. These findings are of considerable interest. A few minor revisions are listed below.
1. The title should be more concise.
2. Lines 33 & 336. 20.0% not 20%.
3. Line 169. Nucleotide sequencing not DNA sequence.
4. Line 172. Dr. not Dra?
5. Lines 195-196. Bp not pb.
6. Fig.1. I can hardly find strain 4385VT_D_BA.
Minor editing of English language required -- e.g., possesses not possess (line 25).
Author Response
"Please see the attachment."

Reviewer 4 Report
Transversal description of diarrheic disease caused by rotavirus A in calves of Argentina with phylogenetic analysis in view of establishing a link with past USA origin and with transmission to Uruguay farms. The amount of molecular results is already very large for this transversal description. It is a first step toward long-time follow up of virus evolution within Argentina in comparison with other countries. The pratical aspects of veterinary vaccine development are not discussed. Maybe a few words in the discussion on this aspect would be interesting for the not specialised reader.
Author Response
"Please see the attachment."
